# Deficits in Cerebellum-Dependent Learning and Cerebellar Morphology in Male and Female BTBR Autism Model Mice

Elizabeth A. Kiffmeyer, Jameson A. Cosgrove, Jenna K. Siganos, Heidi E. Bien, Jade E. Vipond, Karisa R. Vogt and Alexander D. Kloth *

Department of Biology, Augustana University, Sioux Falls, SD 57197, USA
* Correspondence: alexander.kloth@augie.edu

**Abstract:** Recently, there has been increased interest in the role of the cerebellum in autism spectrum disorder (ASD). To better understand the pathophysiological role of the cerebellum in ASD, it is necessary to have a variety of mouse models that have face validity for cerebellar disruption in humans. Here, we add to the literature on the cerebellum in mouse models of autism with the characterization of the cerebellum in the idiopathic BTBR T + Itpr3$^{tf}$/J (BTBR) inbred mouse strain, which has behavioral phenotypes that are reminiscent of ASD in patients. When we examined both male and female BTBR mice in comparison to C57BL/6J (C57) controls, we noted that both sexes of BTBR mice showed motor coordination deficits characteristic of cerebellar dysfunction, but only the male mice showed differences in delay eyeblink conditioning, a cerebellum-dependent learning task that is known to be disrupted in ASD patients. Both male and female BTBR mice showed considerable expansion of, and abnormal foliation in, the cerebellum vermis—including a significant expansion of specific lobules in the anterior cerebellum. In addition, we found a slight but significant decrease in Purkinje cell density in both male and female BTBR mice, irrespective of the lobule. Finally, there was a marked reduction of Purkinje cell dendritic spine density in both male and female BTBR mice. These findings suggest that, for the most part, the BTBR mouse model phenocopies many of the characteristics of the subpopulation of ASD patients that have a hypertrophic cerebellum. We discuss the significance of strain differences in the cerebellum as well as the importance of this first effort to identify both similarities and differences between male and female BTBR mice with regard to the cerebellum.

**Keywords:** autism spectrum disorder; mouse model; idiopathic; cerebellum

## 1. Introduction

Autism spectrum disorder (ASD) is a neurodevelopmental disorder marked by socio-communicative deficits, repetitive behaviors, and stereotyped interests [1]. It is commonly associated with several neurological and non-neurological comorbidities, including motor delay and disruption, cognitive delay, epileptic seizures, and gastrointestinal disturbances [1]. It is estimated that 1 in 44 children born today will receive a diagnosis of ASD, with males being 3–4.2 times more likely to receive an ASD diagnosis than females [2–4], though this number might represent substantial underdiagnosis of girls and women with ASD [2,5,6]. Over the last two decades, there has been an avalanche of research addressing the genetics and neural correlates of ASD, with the long-term goals of identifying biomarkers for early diagnosis and discovering effective treatments for all patients.

The cerebellum has emerged as a brain area of intense interest for ASD researchers, sparked by three lines of evidence that have been reviewed widely [7–13]. First, many ASD patients have cerebellar malformation, including abnormal cerebellar volume [14–17], alteration of Purkinje cell shape and density [18–21], or disruption of cerebellar white matter tracts [22]. This malformation is often observed early on in life and is strongly predictive of a later diagnosis of ASD [23,24]; for this reason, it has been hypothesized

that the cerebellum may play a key early role in the development of brain areas associated with the core ASD behaviors [13]. Second, the cerebellum is an area of the brain in which many ASD susceptibility genes are highly co-expressed, suggesting that mutations at these loci may disrupt cerebellar function [25]. Third, the cerebellum is the locus for disruptions of motor behavior, which are observed in up to 87% of ASD patients [26,27]. Delay eyeblink conditioning, a form of classical conditioning known to require an intact cerebellum [28–30], is also commonly disrupted in ASD patients, who learn the task more slowly, learn to perform the task at a lower rate, or produce inadequate motor responses associated with the task [31–33]. More recent work has gone beyond the motor role of the cerebellum: this work has focused on malformation or malfunction of specific lobules of the cerebellum that are connected with brain areas associated with core ASD behaviors. These studies suggest a regional specificity to disruptions of cerebellar anatomy, activity, and behavior [11,14,15,19,34–36]. Key questions about the relationship between cerebellum and ASD remain—including the exact role of the specific cerebellar lobules in the development of the disorder and the degree to which these findings apply equally to male and female patients—but are beginning to be addressed through preclinical studies, including those employing animal models.

A large fraction of the work on the cerebellum in ASD animal models has focused on rodents modeling single, high-confidence susceptibility genes and environmental models of maternal infection and toxin exposure [37,38]. This work has identified features of the cerebellar pathophysiology that are highly penetrant across ASD cases, uncovering the causative role of region-specific cerebellar function in the development of ASD, an has provided an important proving ground for novel therapeutics that may be used in patients [36,39–50]. In addition, these studies have begun to identify high-confidence targets for rescue in therapeutics studies. For example, it has recently been suggested that delay eyeblink conditioning, which is affected in ASD model mice [45], may be a target that has a one-to-one correspondence between preclinical models and patients [51]. While this work has been critical for understanding the role of the cerebellum in ASD, it has been narrowly focused on models that represent a remarkably small fraction of syndromic or environmental cases in ASD [52]. It remains to be seen how broadly these findings apply to idiopathic cases of ASD, which represent most patients and capture the complex genetic and environmental etiology of the disease. To answer this question, it is important to examine the cerebellum in idiopathic rodent models of ASD.

One commonly used idiopathic mouse model of ASD is the BTBR T + Itpr3$^{tf}$/J (BTBR) mouse [53,54]. This inbred mouse line displays many phenotypes that are analogous to the core disruptions seen with ASD, including disrupted social behavior; disrupted ultrasonic vocalization; deficient performance in cognitive tasks; and repetitive species-specific behaviors such as excessive grooming and disrupted marble burying [55–60]. Few studies have determined whether these phenotypes occur equally in male and female mice [61,62]. Studies that have examined the cerebellum in BTBR mice have found hyperplasia [63,64], disrupted gene expression and epigenetic regulation [65], and signs of immune dysfunction and oxidative stress [65,66]. Only one recent study has examined the BTBR cerebellum on a lobular level, discovering altered neuronal signaling associated with social behavior in lobules IV/V [67]. Despite these findings, there are some significant open questions about the BTBR mouse model, including the degree to which cerebellum-related behavior is dysfunctional, whether disruptions to Purkinje cell density and morphology are present, whether cerebellar effects differ based on lobule, and whether these findings are present in both males and females. Addressing these questions will be important in establishing the BTBR strain as a valid mouse model for exploring the role of the cerebellum in ASD.

In the present study, we examine cerebellum-specific motor learning in the BTBR mice to see if the strain phenocopies what has been observed in ASD patients. We also investigate anatomical and morphological alterations in the adult BTBR mice and determine whether these alterations depend on which lobule of the cerebellar vermis is affected. Importantly,

we examine, for the first time, whether sex is an important biological variable in cerebellar dysfunction in BTBR mice.

## 2. Materials and Methods

### 2.1. Animals

Male and female BTBR T + Itpr3$^{tf}$/J (BTBR) mice were bred at Augustana University using breeding pairs obtained from Jackson Laboratories, Bar Harbor, Maine (stock no. 002282; RRID:MGI:2160299). Male and female C57BL/6J (C57) mice were bred at Augustana University using breeding pairs obtained from Jackson Laboratories, Bar Harbor, Maine (stock no. 000664; RRID:IMSR_JAX:000664). Mice were between 8 and 16 weeks old in all experiments. Sample sizes for each experiment–consistent with prior experiments on the cerebellum in ASD model mice [45]—are shown in Table 1, listed by experiment and figure.

**Table 1.** Sample size for all experiments, listed by sex, strain, experiment, and figure number. Shaded cells are indicated for figures that only report data from a single sex. *, sample size reported as number of cells/number of mice.

| Experiment (Figure) | Male Mice | | Female Mice | |
|---|---|---|---|---|
| | C57 | BTBR | C57 | BTBR |
| Rotarod (Figure 1A,B) | 8 | 8 | 8 | 8 |
| Eyeblink conditioning (Figure 1C,D) | 13 | 12 | 10 | 11 |
| Brain weight (Figure 2A) | 10 | 12 | - | - |
| Vermal anatomy on Nissl-stained tissue (Figure 2C–I) | 8 | 8 | - | - |
| Brain weight (Figure 3A) | - | - | 7 | 10 |
| Vermal anatomy on Nissl-stained tissue (Figure 3C–I) | - | - | 6 | 6 |
| Purkinje cell density (Figure 4B,C) | 8 | 7 | 6 | 6 |
| Golgi–Cox Purkinje cell analysis (Figure 5B–D,F–H) * | 18/6 | 20/7 | 18/5 | 22/6 |
| Golgi–Cox Spine density analysis (Figure 5E,I) * | 27/10 | 25/10 | 15/6 | 15/5 |

All mice were housed on a 12 h light-dark cycle (7 a.m.–7 p.m.) in open-top mouse cages (Ancare, Bellmore, NY, USA) in groups of 2–5 littermates per cage. Animals had ad libitum access to food and water during this period. All procedures were conducted in accordance with protocols approved by the Augustana University Institutional Animal Care and Use Committee.

### 2.2. Accelerating Rotarod

Testing on the accelerating rotarod, which measures motor function and motor coordination [68], was carried out as previously described [69]. Briefly, mice were tested on two separate days, with three trials delivered on the first day and two trials delivered 48 h later. Each day began with 30 min of habituation in a brightly lit room. During each trial, mice were placed using a wooden dowel into one of four lanes of a rod rotating at a constant speed of 4 rpm. Once the trial began, the rotarod accelerated to a speed of 40 rpm over 5 min. The trial for each mouse ended when the mouse fell off the rotarod, completed two complete somersaults around the rotarod, or reached the end of the 5 min trial; end-of-trial times were recorded. Subsequent trials on the same day started 10 min later. The rotarod was cleaned with 70% ethanol between trials.

### 2.3. Surgery

Surgery was conducted according to previously published protocols [45]. Briefly, behavioral mice had a custom titanium head plate surgically attached to their skulls. During surgery, each mouse was anesthetized with isoflurane (1–2% in oxygen, 1 L/min,

for 15–25 min) and mounted in a stereotaxic head holder (David Kopf Instruments, Tujunga, CA, USA). The scalp was shaved and cleaned, and an incision was made down the midline of the scalp. The skull was cleaned, and the margin of the incision was held open using cyanoacrylate glue. The center of the head plate was positioned over bregma and attached to the skull with Metabond dental adhesive (Parkell, Edgewood, NY, USA). Following surgery, the mice were monitored for at least 24 h as they recovered from the surgery.

### 2.4. Eyeblink Conditioning

Eyeblink conditioning experiments were conducted according to previously published protocols (Supplementary Figure S1A) [45]. Briefly, eyeblink conditioning consisted of 3 sessions of habituation followed by 12 sessions of training, with each session taking place in a sound-proof, light-proof box [70]. During each session, animals were head-fixed to a metal support structure and atop a freely rotating foam wheel (constructed from EVA Bumps Foam Roller, 6″ diameter, Bean Products, Chicago, IL, USA). Following the 3 sessions of habituation, animals sat stably and calmly above the wheel, locomoting freely on occasion, without any struggling, as in prior experiments using this technique [45,71,72]. This position allowed a platform for delivering unconditioned (US) and conditioned (CS) stimuli to the animal in a controlled manner. The US (airpuff, 30–40 psi) could be delivered to the cornea through a P1000 pipette tip. The intensity and timing of the puff were controlled by a Picospritzer III (Parker Hannefin, Lakeview, MN, RRID:SCR_018152) connected to a compressed air tank. The position of the needle was adjusted each day for each mouse to ensure that a complete eyeblink was induced by the airpuff. The conditioned stimulus (CS; ultraviolet LED) was delivered to the contralateral eye. Eyelid deflection was monitored using a PSEye Camera run by custom Python software (RRID:SCR_008394) [71]. This same software automatically initiated the trials and delivered the US and the CS via a digital-analog conversion board (National Instruments, Austin, TX, USA). No measurements were taken from the foam wheel.

The animals were allowed to habituate to this apparatus for at least 120 min total over the course of 3 days. Over this time period, the animals demonstrated that they could run freely on the wheel without struggling. Following habituation, acquisition took place over 12 training sessions (1 session/day, 6 days/week), during which the animals received 22 blocks of 10 trials each. The CS (ultraviolet light, 280 ms) was paired with an aversive US (airpuff to the cornea, 30–40 psi, 30 ms, co-terminating with the CS). Each block consisted of 9 paired US-CS trials and 1 unpaired CS trial, arranged pseudorandomly within the block. Each trial was separated by a randomly assigned interval of at least 12 s.

Videos were then analyzed offline using a custom MATLAB (Mathworks, Natick, MA, RRID:SCR_001622) script with experimenter supervision (Supplementary Figure S1B–D) using a method similar to that previously published [73]. Regions of interest containing the eye receiving the corneal airpuff (contralateral to the eye receiving the CS) and part of the animals' faces were smoothed, thresholded, and binarized. Then, the number of white pixels—corresponding to total eyelid closure—was tracked across every frame of the video. For each US-CS trial, data within 1500 ms of the recorded US onset was normalized to the range between the signal minimum during the 280 ms period following CS onset and the signal maximum during the 500 ms period following US onset. Consistent with the prior literature [45], a successful conditioned response (CR) occurred on a US-CS trial if the normalized signal exceeded 0.15 between 100 and 250 ms following CS onset; a trial was excluded if the normalized signal exceeded 0.15 prior to this period. Data are reported as percent CR performance, the percentage of counted trials on which a successful CR occurred. For each unpaired CS trial, the recorded response was normalized to the size of the UR during the previous 9 US-CS trials, then evaluated for the presence of a CR; a CR was counted as present if it exceeded 0.15 between 100 ms and 400 ms after the onset of the CS and remained below 0.05 below 0 ms and 99 ms. Peak time was calculated from smoothed CS curves and averaged across the final three training sessions for each animal.

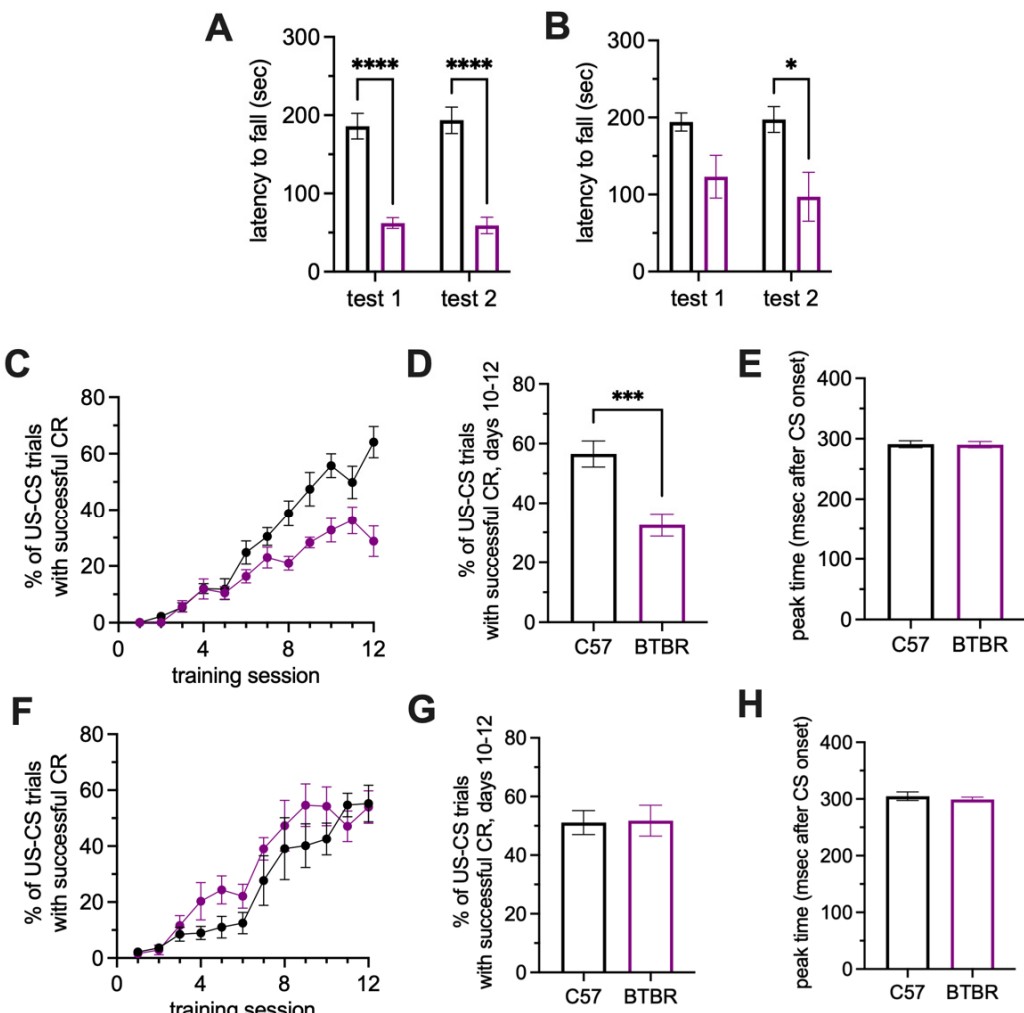

**Figure 1.** Male BTBR show motor learning and coordination deficits, while female BTBR mice show only motor coordination deficits. (**A**) Male BTBR mice fall earlier than male C57 mice across two days of rotarod testing. (**B**) Female BTBR mice fall earlier than female C57 mice across two days of rotarod testing. (**C**–**E**) Male BTBR mice lag behind C57 mice in conditioned response performance in the delay eyeblink conditioning task across twelve training days (**C**) with a significant difference at the end of training (**D**). However, there is no difference in response timing on CS trials (**E**). (**F**–**H**) Female BTBR mice reach comparable levels of conditioned response performance in the delay eyeblink conditioning task across twelve training days (**F**) with no difference in performance at the end of training (**G**). However, there is no difference in response timing on CS trials (**H**). Black, C57B6/J mice; purple, BTBR mice. Error bars denote standard error of the mean. Asterisks denote significant results from two-sample *t*-tests (**A**,**B**,**D**,**F**) or planned comparisons following significant effects in a two-way ANOVA (**C**,**E**). *, $p < 0.05$; ***, $p < 0.001$; ****, $p < 0.0001$.

## 2.5. Tissue Processing and Analysis

Tissues from BTBR and C57 mice were used to analyze the cerebellum at the gross anatomical and cellular levels. All experiments were conducted using previously published protocols [45] but will be recapitulated here. For Nissl staining and immunohistochemistry, mice were anesthetized with 0.15 mL ketamine-xylazine (0.12 mL 100 mg/mL ketamine and 0.80 mL mg/mL xylazine diluted 5× in saline), weighed, and transcardially perfused with 4% formalin in pH 7.4 phosphate-buffered saline (PBS). The brain was extracted, weighed, and stored at 4 °C in 4% formalin in PBS for 24 h. Thereafter, brains were stored in 0.1% sodium azide PBS at 4 °C for vibratome sectioning. For Golgi–Cox staining, mice were deeply anesthetized with gaseous isoflurane and decapitated immediately. The brain

was removed quickly into ice-cold PBS and processed using the FD Rapid GolgiStain kit (FDNeurotechnologies, Inc., Columbia, MD, USA) according to manufacturer instructions.

For Nissl staining, the cerebellum was sliced sagittally into 50 μm sections with a vibrating microtome (Compresstome, Precisionary Instruments, Greenville, NC, USA; RRID:SCR_018452). Every fourth section slice was mounted onto gelatinized Fisherbrand SuperFrost microscope slides (Thermo Fisher Scientific, Waltham, MA, USA) and allowed to dry overnight before being stained. Other sections were stored in PBS for immunohistochemistry (see below). Standard Nissl stain procedures were used as previously published [45], and the slides were sealed and coverslipped with Permount (Fisher Scientific, Fair Lawn, NJ, USA) before being imaged with 4× objective and 10× eyepiece magnification on a Leica LSI 3000 microscope. Serial images based on Allen Mouse Brain Reference Atlas-referenced sections (RRID:SCR_013286) were taken from vermal (sections 10–11); these locations were approximately 1000–1100 μm apart [74,75]. From these images, we measured the length of the molecular and granule cell layers in each section, the area of the molecular and granule cell layers in each section, and the overall section areas. Thickness was determined using a previously published technique [44]. We also counted the number of lobules in each section. All image analysis took place using ImageJ (National Institutes of Health, Bethesda, MD, RRID:SCR_003070).

For immunohistochemistry, the cerebellum was sliced sagittally into 50 μm sections with a Compresstome and stored in PBS. Sections were immunostained with goat anti-calbindin (1:300) as the primary antibody and anti-goat Alexa Fluor 488 (1:200) as the secondary antibody (Invitrogen, Eugene, OR, USA). Sections were counterstained with 4′,6-diamidino-2-phenylindole (DAPI, 1:1000; Invitrogen, Eugene, OR). The sections were mounted onto gelatinized slides and left to dry (at least 2 h) before being coverslipped with VectaShield (Vector Laboratories, Burlingame, CA, USA). The sections were imaged with 10× objective and 10× eyepiece magnification on a Leica LSI 3000 microscope. Purkinje cell density was measured in medial and lateral sections on a lobular basis by measuring the length of the cell layer and counting the number of calbindin-positive cells in each lobule using ImageJ.

For Golgi–Cox staining, the cerebellum was sliced sagittally in 120 μm sections using a Compresstome. The sections were mounted on slides and dried overnight in darkness before being processed according to the FD Rapid GolgiStain kit instructions. After processing, slides were coverslipped and sealed with Permount. The sections were then imaged with 40× and 100× objectives and a 10× eyepiece on a Leica LSI 3000 microscope. The maximum height of the dendritic arbor and the cross-sectional area of the soma was measured using ImageJ. In addition, Sholl analysis was conducted on images taken at 20× objective and 10× eyepiece magnification to quantify the complexity of the dendritic arbor. Briefly, the number of intersections of the dendritic arbor with concentric circles drawn using ImageJ at 8 μm intervals from the soma was counted [76]. In addition, the dendritic spine density for these cells was quantified from the distal dendrites in an unbiased manner. Each cell was examined with 100× oil-immersion objective and 10× eyepiece magnification, the spines on every seventh branchlet (with a random starting point) were counted, and the length of the branchlet was measured. Density was calculated by dividing by the length of the branchlet.

### 2.6. Statistics

All histological data was collected by experimenters blinded to the mouse strain. Behavioral data could not be collected by blinded experiments because of the coat color of the mouse; however, the data collected from these experiments were either processed in a semi-supervised manner or statistically analyzed by an experimenter blinded to mouse strain. Statistical tests used in each experiment are summarized in Supplementary Materials Excel File. Eyeblink conditioning data were analyzed using two-way ANOVAs with repeated measures; main strain effects were reported regardless of significance, whereas main session effects (which would indicate learning over time) are significant, and session × strain interactions are not

significant unless otherwise indicated. Two-way ANOVA tests with Bonferroni-corrected post hoc comparisons were used for comparing layer and lobule area thickness and Purkinje cell density in the cerebellum; main strain effects were reported regardless of significance. Two-way repeated measures ANOVA tests with Bonferroni-corrected post hoc comparisons were used for data from the Sholl analysis. All pairwise statistical tests were unpaired two-sample t-tests unless otherwise noted. The data were analyzed using Prism (GraphPad Software, San Diego, CA, RRID:SCR_002798). The significance level was $\alpha = 0.05$ unless otherwise noted. All results are depicted as mean $\pm$ standard error of the mean (SEM) unless otherwise noted.

### 2.7. Code Availability

All code used in data collection and analysis is available upon request.

### 3. Results

We carried out four sets of experiments to uncover strain differences that depended on sex between C57 and BTBR mice. In describing the results below, we report precise *p*-values; exact statistics can be found in Supplementary Materials Excel File.

In order to identify potential disruptions of motor coordination, we carried out the accelerating rotarod task on BTBR mice and C57 controls over the course of two training days. Male BTBR mice fell off the accelerating rotarod significantly earlier than their C57 controls on both days (Figure 1A; main effect of strain, $p < 0.0001$; differences on both days, $p < 0.0001$), indicating a severe inability to adapt to new motor circumstances. Likewise, female BTBR mice tended to fall off the accelerating rotarod significantly earlier than their C57 controls, particularly on testing day 2 (Figure 1B; main effect of strain, $p = 0.0103$; test day 1, $p = 0.0824$; test day 2, $p = 0.0105$).

We also tested whether BTBR mice were deficient in delay eyeblink conditioning, a motor learning task known to require the cerebellum [28–30,77]. Over the course of 12 training sessions, male BTBR mice lagged significantly behind their C57 counterparts in terms of conditioned response performance (Figure 1C; training session $\times$ strain interaction, $p < 0.0001$), particularly on days 8, 10, and 12 (Bonferroni corrected post hoc tests, $p < 0.05$). When we examined the average conditioned response performance rate over the last three days of training, conditioned response rates in male BTBR mice were significantly lower than those of C57 male mice (Figure 1D; $p = 0.0004$). When we examined successful CS-only trials to determine whether the peak time was altered between strains, we found no significant difference (Figure 1E; $p = 0.9198$). Intriguingly, when we performed the same experiment over 12 training sessions in female mice, we found no significant difference in the time course of learning between female BTBR mice and their C57 counterparts (Figure 1F; main effect of session, $p < 0.0001$). A small difference between strains, which failed to reach significance, suggested accelerated learning in female BTBR mice, in stark contrast to the lag in the male BTBR mice (main effect of strain, $p = 0.2531$). Expectedly, when we examined the average conditioned response performance rate over the last three days of training, female BTBR mice performed at statistically equivalent levels as female C57 mice (Figure 1G; $p = 0.4743$). When we examined successful CS-only trials to determine whether the peak time was altered between strains, we found no significant difference (Figure 1H; $p = 0.5155$).

We proceeded to examine whether there were differences in cerebellar anatomy between strains. When we examined overall brain weight in male mice, we found no significant difference between strains (Figure 2A; $p = 0.5091$). We then examined Nissl-stained sagittal midline vermal sections of the cerebellum. We noted qualitatively that sections from male BTBR mice tended to be larger than sections from their C57 counterparts and showed signs of abnormal foliation (Figure 2B). When we quantified these differences, we found that vermal sections from male BTBR mice were indeed hyperplastic in terms of overall area (Figure 2C; $p = 0.0004$), with significant expansion across layers of the cerebellum (Figure 2D; main effect of strain, $p < 0.0001$; main effect of layer, $p < 0.0001$),

specifically in the molecular cell layer (MCL; $p = 0.0004$) and granule cell layer (GCL; $p = 0.0098$) but not white matter ($p = 0.2697$). In addition, there was a significant abnormal foliation (Figure 2E; $p < 0.0001$), with the average male BTBR section having four additional folia. We then sought to determine whether the expansion and abnormal foliation were uniform across the cerebellum or whether it varied by lobule. Our analysis confirmed overall expansion across lobules (Figure 2F; main effect of strain, $p < 0.0001$; main effect of the lobule, $p < 0.0001$), while Bonferroni-corrected post hoc tests revealed significant expansion in lobules I/II ($p = 0.0012$), IV/V ($p = 0.0037$), and IX ($p = 0.0396$). Given these results, we tested whether the area occupied by the MCL and GCL varied by lobule. We discovered significant differences between strains for MCL (Figure 2G; main effect of strain, $p < 0.0001$) and GCL (Figure 2H; main effect of strain, $p < 0.001$), with differences appearing largely in the anterior cerebellum. We observed significant differences in both layers in lobules I/II (MCL, $p = 0.0015$; GCL, $p = 0.0076$) and significant differences in GCL in lobules IV/V, VI, and VII. We asked whether the increases we observed were driven by differences in the thickness of the layer rather than an increase in the perimeter of the section and found no significant effect of strain on thickness (Supplementary Figure S2A–C, $p > 0.05$ for main effects of strain). Finally, our analysis confirmed abnormal foliation between strains that depend on lobule (Figure 2I; strain x lobule interaction, $p < 0.0001$); male BTBR mice showed additional folia predominantly in lobules in the anterior cerebellum, including lobules I/II ($p = 0.0076$) and IV/V ($p < 0.0001$), along with lobules VI ($p < 0.0001$) and VII ($p = 0.0004$).

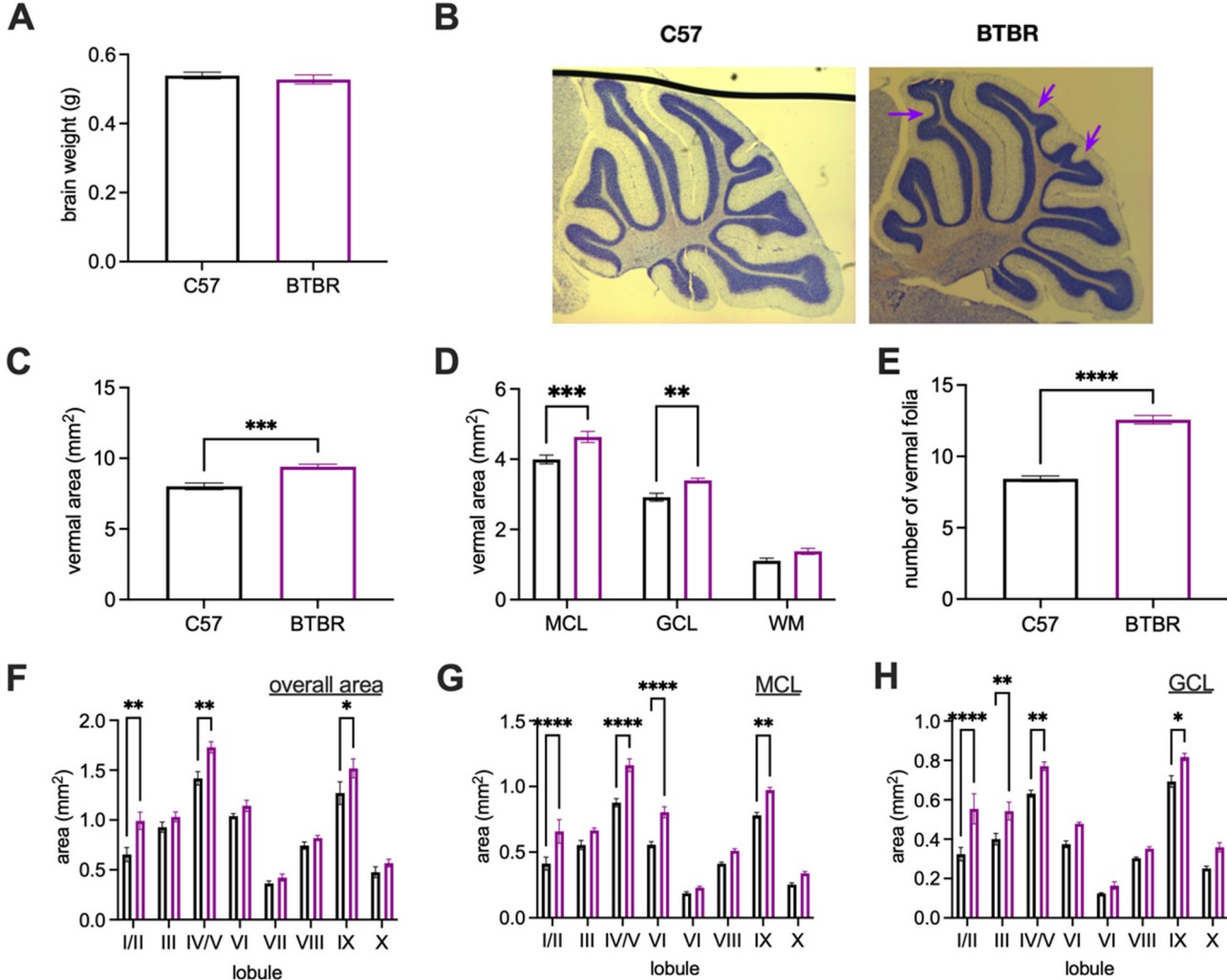

**Figure 2.** *Cont.*

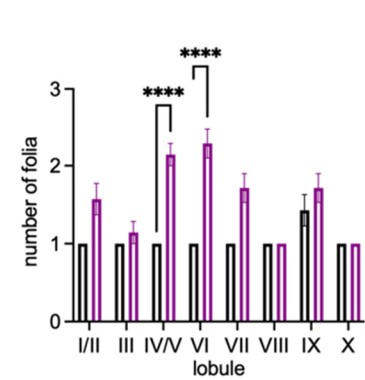

**Figure 2.** Male BTBR mice show vermal enlargement and foliation that varies by lobule. (**A**) Brain weight is comparable between strains. (**B**) Representative image of gross anatomical differences between C57 (left) and BTBR (right) sagittal vermis sections. Arrows identify additional lobules in the BTBR section. (**C**) Area of the midline vermal section is significantly larger in BTBR mice. (**D**) Molecular cell layer (MCL) and granule cell layer (GCL) are significantly enlarged in the BTBR vermis. (**E**) The number of folia in the vermis is significantly different in BTBR mice. (**F**) Enlargement of vermis area in BTBR mice depends on lobule. (**G**) Enlargement of area of the molecular cell layer in BTBR mice depends on lobule. (**H**) Enlargement of area of the granule cell layer in BTBR mice depends on lobule. (**I**) Abnormal foliation in BTBR mice depends on lobule. Black, C57B6/J mice; purple, BTBR mice. Error bars denote standard error of the mean. Asterisks denote significant results from two-sample *t*-tests (**A**–**E**) or planned comparisons following a significant two-way ANOVA (**F**–**I**). *, $p < 0.05$; **, $p < 0.01$; ***, $p < 0.001$; ****, $p < 0.0001$.

In female mice, we first found no significant difference in brain weight between strains (Figure 3A; $p = 0.8221$). As in male BTBR mice, examination of Nissl-stained sagittal midline vermal sections appeared larger and tended to have more folia than sections from their C57 counterparts (Figure 3B). When we quantified these differences, we found that vermal sections were indeed hyperplastic (Figure 3C; $p < 0.0001$), with significant expansion across layers (Figure 3D; strain x layer interaction, $p < 0.0001$). The magnitude of this expansion depended on the layer, with a substantial expansion in the MCL and GCL (Bonferroni-corrected post hoc test, $p < 0.05$). In addition, there was significant abnormal foliation (Figure 3E, $p < 0.0001$), with midline sections from female BTBR mice having, on average, three additional folia than their C57 counterparts. When we examined expansion on a lobule-by-lobule basis, we found that expansion depended on lobule (Figure 3F; strain x lobule interaction, $p = 0.0058$), with Bonferroni-corrected post hoc tests revealing a significant expansion in lobules I/II ($p < 0.0001$), III ($p = 0.0311$), IV/V ($p < 0.0001$), VI ($p = 0.0008$), and IX ($p = 0.0033$). Given these results, we tested whether the area occupied by the MCL and GCL varied by lobule. We discovered significant differences between strains for MCL (Figure 3G; main effect of strain, $p < 0.0001$) and GCL (Figure 3H; main effect of strain, $p < 0.001$), with differences appearing largely in the anterior cerebellum. We observed significant differences in both layers in lobules I/II (MCL, $p < 0.0001$; GCL, $p < 0.0001$) and IV/V (MCL, $p = 0.0090$; GCL, $p = 0.0008$) and other significant differences on one area in lobules III (GCL, $p = 0.0073$) and VI (MCL, $p < 0.0001$). We asked whether the increases we observed were driven by differences in the thickness of the layer rather than an increase in the perimeter of the section and found a significant effect of strain on MCL thickness ($p = 0.0010$) and no significant effect of strain on GCL thickness (Supplementary Figure S2D–F, $p = 0.088$). Finally, our analysis confirmed abnormal foliation between strains that depend on lobule (Figure 3I); female BTBR mice showed additional folia predominantly in lobules in the anterior cerebellum, including lobules I/II ($p < 0.0001$) and IV/V ($p = 0.0008$) as well as lobule VI ($p < 0.0001$).

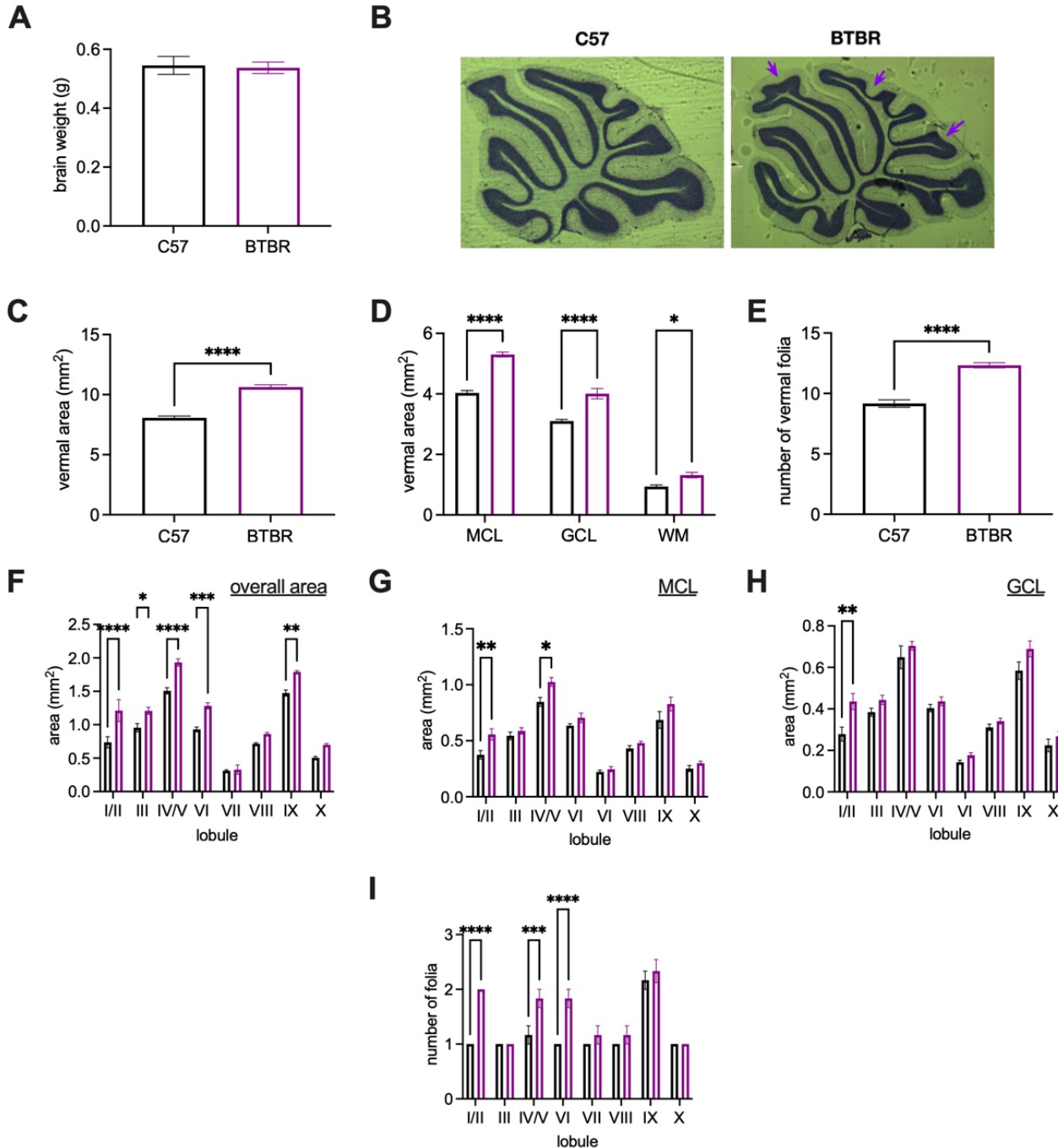

**Figure 3.** Female BTBR mice show vermal enlargement and foliation that varies by lobule. (**A**) Brain weight is comparable between strains. (**B**) Representative image of gross anatomical differences between C57 (left) and BTBR (right) sagittal vermis sections. Arrows identify additional lobules in the BTBR section. (**C**) Area of the midline vermal section is significantly larger in BTBR mice. (**D**) Molecular cell layer (MCL), granule cell layer (GCL), and white matter areas are all significantly enlarged in the BTBR vermis. (**E**) The number of folia in the vermis is significantly different in BTBR mice. (**F**) Enlargement of vermis area in BTBR mice depends on lobule. (**G**) Enlargement of area of the molecular cell layer in BTBR mice depends on lobule. (**H**) Enlargement of area of the granule cell layer in BTBR mice depends on lobule. (**I**) Abnormal foliation in BTBR mice depends on lobule. Black, C57B6/J mice; purple, BTBR mice. Error bars denote standard error of the mean. Asterisks denote significant results from two-sample $t$-tests (**A**–**E**) or planned comparisons following a significant two-way ANOVA (**F**–**I**). *, $p < 0.05$; **, $p < 0.01$; ***, $p < 0.001$; ****, $p < 0.0001$.

We then asked whether the gross anatomical differences were accompanied by cellular differences commonly observed in the cerebellum of ASD patients and autism mouse models, including altered Purkinje cell density and morphology [19,78–81]. An analysis of the linear density of calbindin-positive neurons in midline vermal sagittal sections of male BTBR mice and their C57 counterparts (Figure 4A) showed significant differences between strains (Figure 4B; main effect of strain, $p = 0.0363$); however, Bonferroni-corrected post hoc tests revealed no significant differences with specific lobules ($p > 0.05$ for all comparisons). We performed a similar analysis of the linear density of calbindin-positive neurons in midline sagittal sections of female BTBR mice and their C57 counterparts. As in the male BTBR and C57 mice, there was a significant difference between female BTBR and C57 mice (Figure 4C; main effect of strain, $p = 0.0094$; main effect of the lobule, $p = 0.0046$). At the same time, Bonferroni-corrected post hoc tests also revealed no significant differences with specific lobules ($p > 0.05$ for all comparisons; one near-significant finding in lobule IX).

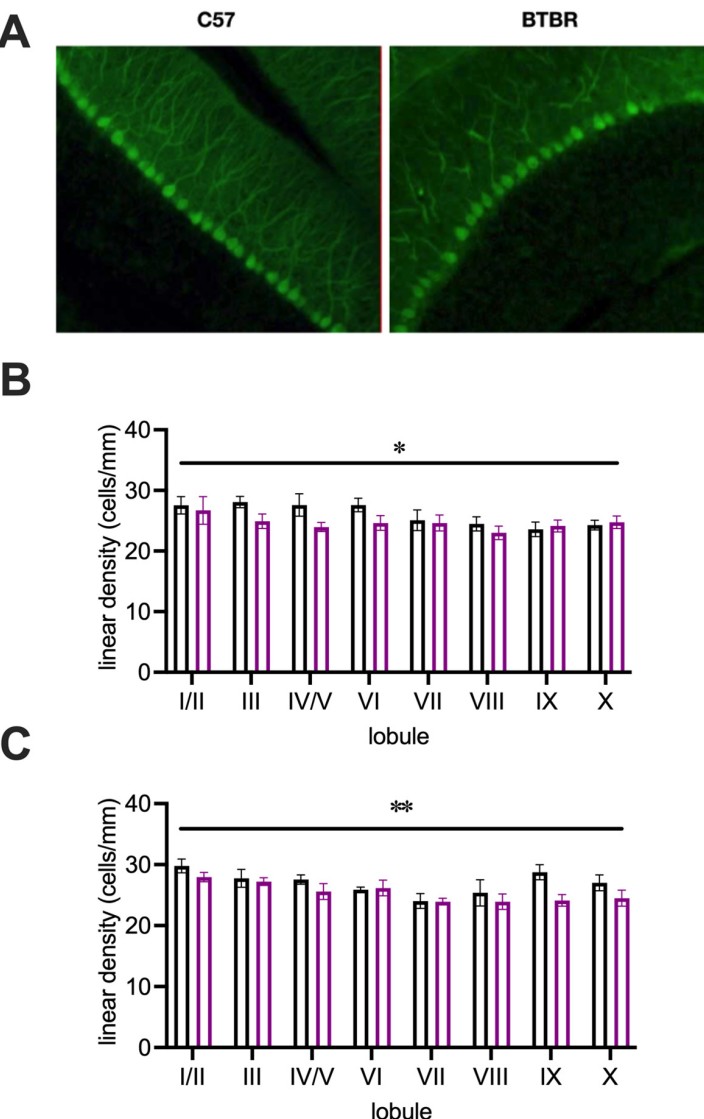

**Figure 4.** BTBR mice of both sexes have slight, global decreases in vermal Purkinje cell density. (**A**) Representative images of calbindin-stained Purkinje cells in male BTBR and C57 mice. (**B**) Lobule-by-lobule analysis shows a broad decrease in male BTBR mice that is not lobule-specific. (**C**) Lobule-by-lobule analysis shows a road decrease in female BTBR mice that is not lobule-specific. Black, C57B6/J mice; purple, BTBR mice. Error bars denote standard error of the mean. Asterisks denote main effect of strain. *, $p < 0.05$; **, $p < 0.01$.

Purkinje cells were also analyzed in terms of cell body size, dendritic arbor height, dendritic spine density, and branching (Figure 5A). When we examined Golgi-stained cells from male mice, Sholl analysis revealed no significant difference in the complexity of the dendritic arbors of Purkinje cells from BTBR and C57 mice (Figure 5B; main effect of strain, $p = 0.6478$). In addition, we found no significant difference in Purkinje cell body size (Figure 5C; $p = 0.2075$) or Purkinje cell dendritic arbor height (Figure 5D; $p = 0.6305$). When we examined differences in dendritic spines on distal branches of Purkinje cells, we identified a trend toward lower dendritic spine density in male BTBR mice (Figure 5E; $p = 0.1478$). When we examined Golgi-stained cells from female mice, Sholl analysis revealed a significantly more complex dendritic arbor in Purkinje cells from female BTBR mice compared to female C57 mice (Figure 5F; main effect of strain, $p = 0.0159$). In addition, there was a trend toward enlarged cell bodies in Purkinje cells from female BTBR mice (Figure 5G; $p = 0.0652$) but no significant difference in Purkinje cell dendritic arbor height (Figure 5H; $p = 0.2261$). Finally, when we examined differences in dendritic spines on distal branches of the Purkinje cells, we identified a significantly lower dendritic spine density in female BTBR mice (Figure 5I; $p < 0.0001$).

Finally, we examined the male and female datasets side-by-side to identify consistent differences among sex and to determine if there were any instances in which the female BTBR mice differed from both male and female C57 mice (Supplementary Figure S3). In most instances, carrying out a comparison via two-way ANOVA revealed the same statistical differences between BTBR and C57 mice in both strains. We verified significant main effects of strain with no significant effect of sex in rotarod performance (Supplementary Figure S3A,B), the number of vermal folia (Supplementary Figure S3G), overall linear density (Supplementary Figure S3H), and dendritic spine density (Supplementary Figure S3K) (all main effects, $p < 0.05$). Likewise, there was no main effect of strain or sex for brain weight (Supplementary Figure S3D, $p > 0.05$). There was one instance in which there was a main effect of sex and strain with no interaction: dendritic spine density (Supplementary Figure S3K, $p < 0.05$). There was a significant sex x strain interaction for eyeblink conditioning performance, consistent with our prior findings (Supplementary Figure S3C, $p = 0.0082$); in this case, BTBR females performed as well as both C57 males and C57 females. There was also significant sex x strain interactions for vermal area (Supplementary Figure S3F, $p = 0.0077$) and Purkinje cell soma area (Supplementary Figure S3I, $p = 0.0146$), suggesting instances where enlargement occurs for one sex or strain group (in Figure 3F, BTBR females have a larger vermal area than all other groups, $p < 0.05$ for all comparisons; and in Figure 3I, the BTBR females have a larger somatic area, $p < 0.05$ for all comparisons). In one instance where the female BTBR mice were statistically different from the female C57 mice, the female BTBR were not quite significantly different from the C57 males, namely in session 2 rotarod ($p = 0.0975$, Supplementary Figure S3B). In two instances, there was a difference between sex that did not appear for both strains: for BTBR vermal area ($p = 0.0015$, Supplementary Figure S3F) and in eyeblink conditioning performance on the last three days ($p = 0.0225$, Supplementary Figure S3C). Overall, with a few exceptions, these comparisons confirm widespread strain differences with few sex differences, with female BTBR mice performing differently from both sexes of C57 mice in both cases.

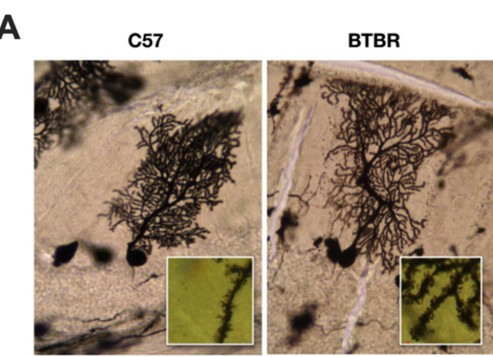

**Figure 5.** *Cont.*

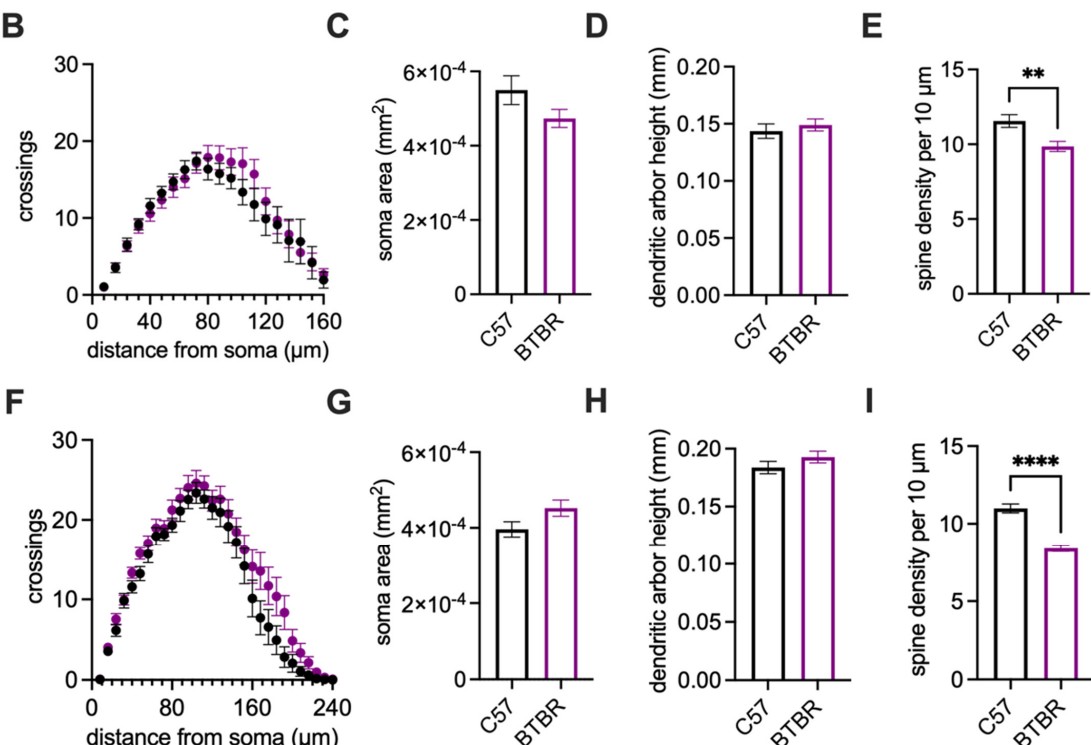

**Figure 5.** Male and Female BTBR mice show alterations to Purkinje cell dendritic branching and spine density. (**A**) Representative examples of Purkinje cells from BTBR (left) and C57 (right) mice. (**B**) Sholl analysis shows no difference between male BTBR and C57 mice. (**C**) Purkinje cell bodies are similar in area in male BTBR mice. (**D**) Dendritic arbor height is not different between groups of male mice. (**E**) Male BTBR mice have fewer dendritic spines on their distal branches. (**F**) Sholl analysis shows a slight increase in the complexity of dendritic arbors of Purkinje cells from female BTBR mice. (**G**) Purkinje cell bodies are similar in area in female BTBR mice. (**H**) Dendritic arbor height is not different between groups of female mice (**I**) Female BTBR mice have fewer dendritic spines on their distal branches. Black, C57B6/J mice; purple, BTBR mice. Error bars denote standard error of the mean. Asterisks denote significant results from two-sample *t*-tests (**C**–**E**) or planned comparisons following a significant two-way ANOVA (**B**). **, $p < 0.01$; ****, $p < 0.0001$.

## 4. Discussion

We set out to characterize cerebellum-specific phenotypes of BTBR T + Itpr3$^{tf}$/J to determine whether it would be a suitable mouse model for understanding the cerebellar basis of ASD in both sexes. We discovered a high degree of concordance between sexes in our measurements, with a small number of exceptions. BTBR mice tend to show deficits in motor learning, with male mice in particular lagging behind in both tasks we examined. At a gross anatomical level, the BTBR cerebellum is hyperplastic, with significant vermal expansion and abnormal foliation occurring most substantially in lobules and IV/V and VI. Purkinje cells tend to have a lower density across the BTBR vermis than their C57 counterparts, though this decrease is not confined to a single lobule. In addition, there are notable disruptions in the structure of the dendritic arbor: BTBR cells most notably have a significantly lower dendritic spine density than C57 cells.

Our finding of significant motor learning impairments in the BTBR mouse model is consistent with previous literature. One prior work by Xiao and colleagues noted a disruption of rotarod performance in male BTBR mice [82]. The present study confirms that finding, while also adding that female mice have a similar—albeit less severe—deficit. Our finding that male BTBR mice have a deficit in delay eyeblink conditioning, a motor task known to require the cerebellum, is novel but consistent with other ASD mouse models. Prior studies show that delay eyeblink conditioning dysfunction is widespread in ASD

mouse models, with deficits in either the ability to acquire delay eyeblink conditioning or to perform the conditioned eyeblink with the correct magnitude or timing [45,47,48,83,84]. The present study adds to this body of the literature. Deficits in eyeblink conditioning tend to cluster with the part of the cerebellar circuit in the eyeblink region that is most likely to be disrupted, setting up future research probing the BTBR cerebellum at the neural circuit level [45]. Delay eyeblink conditioning deficits do arise in ASD patients, with fewer disruptions in the ability to learn and more frequent disruptions in the timing of the conditioned response than is demonstrated here [31–33].

Interestingly, we did not discover the same conditioning deficit in the female mice, and there is some evidence to indicate that female mice acquired conditioning somewhat more quickly than their C57 counterparts. This intriguing finding does mirror a result in the patient literature suggesting faster learning in the delay eyeblink conditioning task [32] (but notice the lack of timing deficits here) and generally mirrors sex differences in the task in the neurotypical population [85]. How might a sex difference in delay eyeblink conditioning arise? Differences in the speed of eyeblink acquisition have been ascribed to the role of the hormonal stress response in learning in female mice [86] or differences in the activity of neurons in the motor areas of the cerebellum [87]. It is possible that sex differences in stress processing [88] or sex differences in the electrophysiology of Purkinje cells [89] in the BTBR mice might account for this difference. Some researchers have suggested that delay eyeblink conditioning could represent a rare phenotype that occurs similarly in patients and model mice; such a finding would provide easily interpretable outcomes for therapeutic studies and provide a clearer path to understanding the cerebellar pathophysiology of autism [51,90]. However, for eyeblink conditioning to be a useful biomarker, much more work will need to be performed to determine how well mouse models, like our BTBR mouse model, map onto a segment of the patient population in males and females.

We discovered that mice of both sexes have vermal hyperplasia and abnormal foliation. The finding that male mice have hyperplasia is consistent with previous studies showing that the cerebellum occupies a larger percentage of brain volume in BTBR mice than it does in C57 mice [63,64]. Our finding that the same feature occurs in female mice is novel. In addition, we are the first group to uncover hyperplasia that is regionally specific, identifying significant enlargement in the anterior cerebellum, lobule VI, and lobule IX. This finding of vermal hyperplasia is certainly at odds with literature that shows that many ASD patients have cerebellar hypoplasia [15,23,91], though there are reports that are more consistent with our findings of regional hyperplasia [14,16,92]. Indeed, an exhaustive study of twenty-six ASD mouse models suggests that malformation of the cerebellum varies widely and may be indicative of multiple subpopulations within the ASD patient population [63]. Our findings may apply more narrowly to one of these subpopulations. In addition, we have discovered abnormal foliation in both male and female mice, confirming one previous study in male mice [82] and notable because the earlier literature had rejected the notion of anatomical abnormality in the cerebellum [93]. This foliation defect is indicative of disruption of the maturation of the cerebellum early in postnatal murine brain development [94]; the observation that early granule cell layer development as well as Purkinje cell migration defects [82] might account for this disrupted foliation is in line with our observation of regional differences in the area of the granule cell layer and molecular cell layer. However, further investigation is required to understand the significance of hyperfoliation of some lobules and not others.

We also discovered a disruption of Purkinje cell density and morphology in both male and female mice. Our finding of a reduced Purkinje cell density is consistent with patient reports of lowered Purkinje cell number [19,21,80,81] and is consistent with a report on reduced Purkinje cell number in juvenile BTBR male mice [82]. The same finding was observed in female mice for the first time. This finding is consistent with the idea that deficits in Purkinje cell migration during late cerebellar development might underlie the anatomical differences in the cerebellum [82]. However, unlike our other measurements, one previous study in BTBR male mice [82], and one notable study showing regional

specificity in Purkinje cell density loss [19], we did not find evidence that any one lobule was more disrupted than another. Likewise, we did not find any differences in the size of Purkinje cell bodies as observed in BTBR mice [82] and in patients, and we only observed minor differences in the complexity of the dendritic arbor. Notably, in both male and female BTBR mice, we noted a reduction in the density of dendritic spines, perhaps indicating a reduction of excitatory drive to Purkinje cells that is critical for cerebellar development and cerebellar learning. While this finding is different from that of increased numbers of immature dendritic spines in male juvenile BTBR mice [82], these results from adult mice might indicate an over-pruning process that takes place later in development. However, the significance of dendritic spine density in ASD remains an open question, particularly because of the high variability of the direction and magnitude of dendritic spine dysgenesis across ASD studies [20,95].

Our findings identified lobules I/II, IV/V, VI, and IX as drivers of the differences between the BTBR cerebellum and the C57 cerebellum in both sexes. What is the significance of these lobules in ASD and ASD-related behavior? Dysplasia has been long observed in some lobules but not others [14,15,19], but studies of connectivity have revealed the deeper, nonmotor role of the cerebellum [96]. Such studies have identified the anterior vermis—lobule I through lobule IV/V—as centrally involved in the stereotyped behavior seen in ASD patients because of its functional connectivity with cerebral areas involved in this behavior [34]. Likewise, the posterior vermis—including lobule IX—has been observed to be involved in emotional regulation and social function [34]. Lobule VI has also been identified for its role in stereotyped behavior [97]. Regarding lobule IV/V, a recent study using chemogenetic manipulation in BTBR mice shows a complex role for the lobule in motor function, social behavior, and memory [67]. It is possible that the lobules have a complex relationship with ASD-relevant behavior. However, despite the growing body of evidence illustrating a clear link between cerebellar lobules and specific aspects of ASD behavior, the current study does not attempt to connect the observed regional abnormalities with any individual behavior. Furthermore, this study did not attempt to measure hemispheric areas like crus I and crus II that have been targeted for their involvement in social behavior [36,40]. Making these connections will require further investigation.

Possible limitations to this study include the range of ages of all mice tested and potential confounds from the estrous cycle in female mice, both of which are variables that could ostensibly affect strain or sex differences demonstrated here. In the present study, we use mice between 8 and 16 weeks old, which is in the young adult to adult ranges in terms of mouse age. The age range used here is consistent with previous studies [40,45], but there is reason to suggest that age in young adulthood may affect the stability of some behaviors [98]. In terms of the stability of eyeblink conditioning performance with age, one prior suggests that the behavior is relatively stable between 4 months and 9–12 months of age in C57BL/6 mice [99], while another suggests a wider range of stability for C57BL/6J mice, from 2 months to 10 months [100], within the range present experiment. To date, similar comparisons have not been made in BTBR mice. While other studies have looked at the ontogeny of eyeblink conditioning in very young rodents [101–103], the literature is notably scarce when it comes to the young adult time point (2 months) in mice. Cerebellar anatomy seems similarly stable within this age range [104,105]. On the other hand, rotarod performance may be affected: one prior study suggests small but significant differences between 2–3 and 4–5-month-old C57BL/6J mice [98]. Such a difference might account for the lack of a significant difference in male mice on test day 1, but it would not account for the observed differences on the following test day. In the present study, we also used female mice without monitoring the estrous cycle. Previously regarded as a potential source of variation in measurements in female mice, a widely cited meta-analysis shows that female mice, when tested irrespective of their estrous cycle, showed no significant increase in variation compared to male mice [106]. This lack of difference in variability has recently been confirmed for delay eyeblink conditioning, motor behavior, and other aspects

of cerebellar function in another study [87]. It is possible that other factors that vary with sex that are cited in the Oyaga study, like wheel running [107] and response to stressful and anxiogenic situations [108], may account for the differences noted in our study as well as in the literature. Likewise, rotarod performance [109] and aspects of cerebellar anatomy [110] measured in this study are unlikely to be affected appreciably by the hormonal state of female mice. It is, however, possible that BTBR mice have altered variability in response to the estrous cycle, which has been examined in the occasional study [111] but never with regard to cerebellar behavior. Finally, we should acknowledge that a more rigorous investigation on eyeblink conditioning that looks at different modalities for conditioned stimuli and different delay intervals may reveal substantial differences in learning that are more like those in female mice.

The present study expands the BTBR literature in a few significant ways. First, it highlights ways in which male and female BTBR brains both differ from their C57 counterparts and from each other. The goal of recent pushes in our field to examine sex as a biological variable is justified [112]—it ensures that we do not ignore a significant portion of the patient population. As one of the few studies that have examined both male and female BTBR, the present project asks whether the BTBR mouse model is valid for studying all aspects of ASD in all patients. Future studies should attempt to pinpoint the mechanism underlying the sex differences we have observed here. Second, our study is the first to test whether cerebellum-dependent behaviors—namely, delay eyeblink conditioning—are disrupted in these mice. The study helps put the BTBR mouse model in the larger context of studies in other mouse models that have observed delay eyeblink conditioning deficits as a highly penetrant feature of ASD. Third, our study identifies lobule-specific abnormalities that may correlate with the behavioral profile of the BTBR mouse. Future studies should attempt to identify a causative link between lobule-specific disruption or rescue in the BTBR mouse and alterations of behavior. Finally, this study demonstrates the validity of the BTBR mouse model for understanding cerebellar dysfunction as it mirrors phenotypes in at least a segment of the ASD patient population. Future research should continue to characterize this mouse model for the purposes of identifying effective treatments for and understanding the underlying etiology of ASD in a particular patient subpopulation.

**Supplementary Materials:** The following supporting information can be downloaded at: https://www.mdpi.com/article/10.3390/neurosci3040045/s1, Figure S1: Protocol for collecting and analyzing images from delay eyeblink conditioning sessions; Figure S2: The thicknesses of the molecular and granule cell layers depend on sex and lobule; Figure S3: Side-by-side sex and strain analysis of male and female experiments for BTBR and C57 mice; Supplementary Materials Excel File: Full details of statisticaly analyses performed in this study.

**Author Contributions:** E.A.K.: performed experiments; analyzed data; writing; figure creation; editing. J.A.C.: performed experiments; analyzed data; figure creation; editing. J.K.S.: performed experiments; analyzed data; figure creation; editing. H.E.B.: performed experiments; analyzed data; editing. K.R.V.: performed experiments; designed and built experimental apparatus; analyzed data; editing. J.E.V.: performed experiments; designed and built experimental apparatus; analyzed data; editing. A.D.K.: designed and performed experiments; designed and built experimental apparatus; analyzed data; writing; figure creation; editing; obtained funding. All authors have read and agreed to the published version of the manuscript.

**Funding:** This work is supported by an Institutional Development Award (IDeA) from the National Institute of General Medical Sciences of the National Institutes of Health (P20GM103443), by the National Science Foundation/EPSCoR Award No. IIA-1355423 to the South Dakota Board of Regents and by support from the Augustana University Biology Department Endowment.

**Institutional Review Board Statement:** The animal study protocol was approved by the Institutional Animal Care and Use Committee of Augustana University (protocol number 003, 28 March 2018).

**Informed Consent Statement:** Not applicable.

**Data Availability Statement:** Data from the study are available upon request.

**Acknowledgments:** We would like to acknowledge the work of collecting a small amount of data on this project by Abby Reynen and Christina Pickett. Special thanks to Brenda Rieger and Brian Vander Aarde for animal husbandry and technical assistance. Illustrations in Supplementary Figure S1 and Graphical Abstract are the artistic creations of J.K.S.

**Conflicts of Interest:** The authors have no conflict of interest.

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
