# Peer review of "Deficits in Cerebellum-Dependent Learning and Cerebellar Morphology in Male and Female BTBR Autism Model Mice"

_neurosci, doi:10.3390/neurosci3040045_

Round 1

Reviewer 1 Report

This is a review of the paper “Deficits in cerebellum-dependent learning and cerebellar morphology in male and female BTBR autism model mice” by Kiffmeyer et al. This paper examines the cerebellum in a mouse model of ASD through behavioral tests and morphology. The experiments pair behavioral tasks that are known to be cerebellar dependent with several different morphology assessments of the cerebellum. A strength of the paper is that both sexes were used, but it would be stronger if there were direct comparisons between the sexes and strains together. Unfortunately, significant parts of the methods are missing, making it harder to assess the results. The manuscript needs some heavy editing and addition of experimental details before it can be published.

Abstract line 16: “… both mice showed motor coordination deficits characteristic of cerebellar function, … “ This sentence is confusing. By both mice, do you mean both sexes or both strains? Both are mentioned earlier in the sentence. The motor coordination deficits are characteristic of cerebellar function, shouldn’t that be dysfunction? Please re-write this sentence to clarify the meaning.

Abstract line 25: Sex differences in BTBR mice have been shown before. Do you mean the cerebellar differences?

Introduction

Line36: “… males being 4.2 times more likely to receive an ASD diagnosis than females…” This statement is true but not sufficient, the report it comes from is based solely on 8 year olds. Most females are diagnosed with ASD are done so at a later age than males with ASD. This needs to be qualified because it is becoming more clear that female ASD looks different than male ASD, but it exists and may end up being almost as prevalent. It is important that these studies are done in both sexes. More and more studies use both sexes, more than is given credit in this paper.

Methods:

The specific number of mice (only “at least” in methods) for each test needs to be reported either in the Methods or in the figure legends or on the figures.

Please add the rotarod methods.

Line 150: The stationary, freely rotating foam wheel – the animals could locomote freely throughout the experiment. Does that mean they could run without moving their heads for the air puff? Were any measurements taken from the foam wheel? Did the mice have any trouble staying on the wheel?

Eyeblink conditioning: The graphs (Figure 1C&E) are the % of US-CS trials with a successful CR? What happened on the CS alone trials? There should be 22 of them per training session. From the supplemental figure C&D, it looks like the timing of the CR can be measured. The Sears and Steinmetz paper suggested that people with ASD have no problems learning eyeblink conditioning but have poorly timed CRs. Is it possible to assess that in the female mice (as the female BTBR mice seemed to learn faster than the B6 mice).

Statistics Line 253: The experimenters cannot be blind to the strain during behavior tests as the two strains look very different.

Statistics: There are no genotype differences. Please change genotype to strain.  

Results:

It is customary to include the F value and the degrees of freedom when reporting statistics. Please add them. Also p values are usually < 0.05, < 0.01 or < 0.001.

For all of the analyses, were the males and females analyzed together? It would be interesting to see of the female BTBR mice differed from the male B6 mice in these tests.

Figure 2 and 3 would benefit by more labeling of the graphs. Specifically F-H would be enhanced if the cell layer that is being represented were listed  on each graph.

Author Response

We thank the reviewer for their work on this paper. We hope that the following revisions make our manuscript suitable for publication.

This is a review of the paper “Deficits in cerebellum-dependent learning and cerebellar morphology in male and female BTBR autism model mice” by Kiffmeyer et al. This paper examines the cerebellum in a mouse model of ASD through behavioral tests and morphology. The experiments pair behavioral tasks that are known to be cerebellar dependent with several different morphology assessments of the cerebellum. A strength of the paper is that both sexes were used, but it would be stronger if there were direct comparisons between the sexes and strains together. Unfortunately, significant parts of the methods are missing, making it harder to assess the results. The manuscript needs some heavy editing and addition of experimental details before it can be published.

Abstract line 16: “… both mice showed motor coordination deficits characteristic of cerebellar function, … “ This sentence is confusing. By both mice, do you mean both sexes or both strains? Both are mentioned earlier in the sentence. The motor coordination deficits are characteristic of cerebellar function, shouldn’t that be dysfunction? Please re-write this sentence to clarify the meaning.

We have edited the sentence to clarify the meaning of this line of the abstract per the reviewer’s two comments.

Abstract line 25: Sex differences in BTBR mice have been shown before. Do you mean the cerebellar differences?

We have corrected this statement in the abstract per the reviewer’s comment.

Introduction

Line36: “… males being 4.2 times more likely to receive an ASD diagnosis than females…” This statement is true but not sufficient, the report it comes from is based solely on 8 year olds. Most females are diagnosed with ASD are done so at a later age than males with ASD. This needs to be qualified because it is becoming more clear that female ASD looks different than male ASD, but it exists and may end up being almost as prevalent. It is important that these studies are done in both sexes. More and more studies use both sexes, more than is given credit in this paper.

We have addressed this point by citing recent systematic reviews and meta-analyses that both cite the sex ratio as somewhere between 3-4.2:1 and acknowledge the challenges of diagnosing girls and women with autism spectrum disorder. These citations now appear in this part of the Introduction. 

Methods:

The specific number of mice (only “at least” in methods) for each test needs to be reported either in the Methods or in the figure legends or on the figures.

We have produced a table of these values and included them as Table 1 in the manuscript. Table 1 appears in the Results section.

Please add the rotarod methods.

We have added the rotarod methods as section 2.2.

Line 150: The stationary, freely rotating foam wheel – the animals could locomote freely throughout the experiment. Does that mean they could run without moving their heads for the air puff? Were any measurements taken from the foam wheel? Did the mice have any trouble staying on the wheel?

We thank the reviewer for these comments, which allowed us to clarify the description of the apparatus in this part of the Methods section.

Eyeblink conditioning: The graphs (Figure 1C&E) are the % of US-CS trials with a successful CR? What happened on the CS alone trials? There should be 22 of them per training session. From the supplemental figure C&D, it looks like the timing of the CR can be measured. The Sears and Steinmetz paper suggested that people with ASD have no problems learning eyeblink conditioning but have poorly timed CRs. Is it possible to assess that in the female mice (as the female BTBR mice seemed to learn faster than the B6 mice).

Taking this reviewer response in three parts:

(1) We have modified the y-axis labels in Figures 1 C-F (now Figure 1 C-H) to make this clearer.

(2) We have addressed the use of CS-alone trials for peak and timing analysis in the Methods section.

(3) We have included this timing analysis for male and female BTBR and C57 trials in Figure 1E&H. We have also described these results in the text around Figure 1, and have included the appropriate statistics there and in Supplementary Table 1.

Statistics Line 253: The experimenters cannot be blind to the strain during behavior tests as the two strains look very different.

The reviewer is correct. We corrected the text to indicate that blinding was only possible in experiments (histology, etc.) where experimenters could not observe coat color.

Statistics: There are no genotype differences. Please change genotype to strain.

We changed three instances of “genotype” in the statistics section to “strain”.

Results:

It is customary to include the F value and the degrees of freedom when reporting statistics. Please add them. Also p values are usually < 0.05, < 0.01 or < 0.001.

We thank the reviewer for this comment. Although F values and degrees of freedom often appear directly in the text in many manuscripts, in our experience inclusion of these statistics severely impedes the readability of the manuscript. For this reason, we made the decision to include the complete set of statistical details in Supplementary File 1. Indeed, this custom is becoming more frequent in the literature. To make this clear, we have added a line at the beginning of the results section directing readers to Supplementary File 1 for complete statistical details. Consistent with this decision, we decided to report the precise p-values directly in the text for quick evaluation by readers.

For all of the analyses, were the males and females analyzed together? It would be interesting to see of the female BTBR mice differed from the male B6 mice in these tests.

We have performed this analysis and visualized it in Supplementary Figure 3. In most instances where there were strain differences, female BTBR mice differed from both C57 males and C57 females. We have reported these results at the end of the results section.

Figure 2 and 3 would benefit by more labeling of the graphs. Specifically F-H would be enhanced if the cell layer that is being represented were listed  on each graph.

We added labels for “overall area,” “MCL,” and “GCL” to Figures 2F-H and 3F-H for clarity per the reviewer’s suggestion.

Reviewer 2 Report

The article by Kiffmeyer and co-workers investigates cerebellum-dependent learning and cerebellum morphology in the BTBR mouse model of Autism spectrum disorder. The study is carefully performed and the results are clearly presented and discussed. This work adds to previous studies on this well known mouse line, extending the cerebellum-specific neurobehavioral phenotypes of this mouse line, especially thanks to the inclusion of both sexes. My major worry lies in the large range of ages (8-16 weeks), including both juvenile and adult mice and in the apparent lack of assessement of the estrous phase in female mice. The authors should provide more information about the possible impact of age and estrous confoudings on the data and discuss it in the appropriate manuscript section.

Author Response

We thank the reviewer for their work on this paper. We hope that the following revisions make our manuscript suitable for publication.

The article by Kiffmeyer and co-workers investigates cerebellum-dependent learning and cerebellum morphology in the BTBR mouse model of Autism spectrum disorder. The study is carefully performed and the results are clearly presented and discussed. This work adds to previous studies on this well known mouse line, extending the cerebellum-specific neurobehavioral phenotypes of this mouse line, especially thanks to the inclusion of both sexes. My major worry lies in the large range of ages (8-16 weeks), including both juvenile and adult mice and in the apparent lack of assessement of the estrous phase in female mice. The authors should provide more information about the possible impact of age and estrous confoudings on the data and discuss it in the appropriate manuscript section.

We thank the reviewer for this comment. In response, we have added a new penultimate paragraph to the Discussion section, in which we review the available literature on the stability of the measurements we had in this study to variation in age and estrous cycle. We have also pointed out other experimental variables that may influence the outcome of these experiments in other researchers’ hands.